# Novel Thiochromanone Derivatives Containing a Sulfonyl Hydrazone Moiety: Design, Synthesis, and Bioactivity Evaluation

**DOI:** 10.3390/molecules26102925

**Published:** 2021-05-14

**Authors:** Lu Yu, Jiyan Chi, Lingling Xiao, Jie Li, Zhangfei Tang, Shuming Tan, Pei Li

**Affiliations:** 1School of Liquor and Food Engineering, Guizhou University, Guiyang 550025, China; lyu1@gzu.edu.cn (L.Y.); qq1401064120@126.com (J.C.); an1378386891@163.com (L.X.); sushilee0120@163.com (J.L.); tzf18885247153@163.com (Z.T.); 2Qiandongnan Engineering and Technology Research Center for Comprehensive Utilization of National Medicine, Kaili University, Kaili 556011, China

**Keywords:** thiochromanone, sulfonyl hydrazone, antibacterial activity, antifungal activity

## Abstract

A series of novel thiochromanone derivatives containing a sulfonyl hydrazone moiety were designed and synthesized. Their structures were determined by ^1^H-NMR, ^13^C-NMR, and HRMS. Bioassay results showed that most of the target compounds revealed moderate to good antibacterial activities against *Xanthomonas oryzae* pv. *oryzae*, *Xanthomonas oryzae* pv. *oryzicolaby*, and *Xanthomonas axonopodis* pv. *citri*. Compound **4i** had the best inhibitory activity against *Xanthomonas oryzae* pv. *oryzae*, *Xanthomonas oryzae* pv. *oryzicolaby*, and *Xanthomonas axonopodis* pv. *citri*, with the EC_50_ values of 8.67, 12.65, and 10.62 μg/mL, which were superior to those of Bismerthiazol and Thiodiazole-copper. Meanwhile, bioassay results showed that all of the target compounds proved to have lower antifungal activities against *Sclerotinia sclerotiorum*, *Fusarium oxysporum*, *Gibberella zeae*, *Rhizoctonia solani*, *Verticillium dahlia*, and *Botrytis cinerea* than those of Carbendazim.

## 1. Introduction

Plant bacterial and fungal diseases pose serious threats in agricultural production and cause huge economic losses throughout the world each year [1]. In recent years, crop cultivators continually battle with plant bacterial and fungal diseases that affect their crops. The available traditional pesticides used for plant bacterial and fungal diseases control irrationally posed dangers to living systems, killing not only the target bacteria and fungi, but also affecting beneficial living systems [2]. In order to protect plant bacterial and fungal diseases, commercial agriculture relies heavily on the inputs of chemical pesticides and the resistance of plant bacterial and fungal diseases against pesticides is rapidly becoming a serious problem. Therefore, the development of novel and promising antibacterial and antifungal agents is still an urgent task.

Chromone, a kind of botanical active component with extensive biological activities, is widely found in the secondary metabolites of flowers, roots, stems, and pericarp of many plants [3,4]. As early as the late 19th century, Khellin was extracted from the fruit of the Umbellifera, which was widely distributed in eastern Mediterranean countries, and was used as the first chromone drug in clinical applications [5]. Meanwhile, biological activity results showed that both natural and synthetic chromone compounds had a wide range of biological activities, such as antifungal [6,7], antibacterial [6], anticancer [8], and antiviral [9] activity. Thiochromanone, a kind of chromone compound, is an important substance in the synthesis of various active molecules that are extensively used in the intermediate skeleton of drugs and has attracted more and more attention due to their extensive biological activities, including antiviral [10], antibacterial [11,12], antifungal [11,13,14,15], herbicidal [16,17], and insecticidal [18] activity. Therefore, due to its excellent features of low toxicity and easy to be synthetized and derived, thiochromanone is considered to be a leading compound to develop promising agrochemical candidates, which will become a reality. As an important nitrogen-containing compounds in organic synthesis, sulfonyl hydrazones are widely employed to construct C–C, C–N, and C–S bonds. Compounds with sulfonyl hydrazone structural units present various biological activities, including antifungal [19,20,21], antibacterial [22,23], anticancer [24,25,26], and insecticidal [27] activity. However, there are no reports on the synthesis and bioactivity evaluation of thiochromanone derivatives containing a sulfonyl hydrazone structure.

Motivated by the above-mentioned findings and to discover new active small molecules, in this study, using botanical active component thiochromanone as the lead compound, a series of novel thiochromanone derivatives containing a sulfonyl hydrazone moiety were designed, synthesized, and determined their in vitro antibacterial activities against *Xanthomonas oryzae* pv. *oryzae* (*Xoo*), *Xanthomonas oryzae* pv. *oryzicolaby* (*Xoc*), and *Xanthomonas axonopodis* pv. *citri* (*Xac*), as well as their in vitro antifungal activities against *Sclerotinia sclerotiorum* (*S. sclerotiorum*), *Fusarium oxysporum* (*F. oxysporum*), *Gibberella zeae* (*Gibberella zeae*), *Rhizoctonia solani* (*R. solani*), *Verticillium dahlia* (*V. dahlia*), and *Botrytis cinerea* (*B. cinerea*).

## 2. Results and Discussion

### 2.1. Chemistry

Using 4-substituted thiophenol as the raw materials, as shown in Scheme 1, the target compounds **4a**–**4r** were prepared with the yields of 60.0%–84.6%. In the ^1^H-NMR spectra of **4a**–**4r**, the singlet around *δ* = 10.62–11.02 ppm indicated the presence of the –NH– group. The singlet in *δ* = 170.1–170.8 ppm indicated the presence of a C=O group. The ^1^H-NMR, ^13^C-NMR, and HRMS data and spectra for all the synthesized compounds are shown in the Appendix A.

### 2.2. Biological Evaluations

In this study, the in vitro antibacterial activities of the target compounds **4a**–**4r** against *Xoo*, *Xoc*, and *Xac* at 200 and 100 μg/mL were determined by the turbidimeter test [28] and the results were statistically analyzed and listed in Table 1. Most of the target compounds revealed moderate to good in vitro antibacterial activities against *Xoo*, *Xoc*, and *Xac* at 200 and 100 μg/mL. In particular, compounds **4g**, **4i**, **4j**, and **4l** revealed a 100% inhibition rate against *Xoo* at 200 μg/mL; notably compound **4i** and **4l** still achieved a 100% inhibition rate at 100 μg/mL, which were even better than those of Bismerthiazol and Thiodiazole-copper. Meanwhile, Table 1 showed that most of the target compounds proved to have better in vitro antibacterial activity against *Xoc* at 200 and 100 μg/mL. Among of them, compound **4i** exhibited the best inhibitory activity against *Xoc* at 200 and 100 μg/mL, with the inhibition rates of 96% and 90%, respectively, than those of Bismerthiazol and Thiodiazole-copper. In addition, all the target compounds, except compounds **4n**, **4p**, and **4q**, revealed better inhibitory activity against *Xac* than those of Bismerthiazol and Thiodiazole-copper; notably, compounds **4g**, **4i**, and **4l** revealed a 100% inhibition rate against *Xac* at 200 μg/mL.

Based on preliminary bioactivity results, the EC_50_ values of some of the target compounds against *Xoo*, *Xoc*, and *Xac* were also determined and the results were statistically analyzed and listed in Table 2. As shown in Table 2, compounds **4c**, **4f**, **4g**, **4h**, **4i**, **4j**, and **4l** revealed lower EC_50_ values against *Xoo* (8–32 μg/mL), *Xoc* (12–46 μg/mL), and *Xac* (10–38 μg/mL) than those of Bismerthiazol and Thiodiazole-copper. In particular, compound **4i** showed the best in vitro antibacterial activities against *Xoo*, *Xoc*, and *Xac*, with EC_50_ values of 8, 12 and 10 μg/mL, which were superior to those of Bismerthiazol and Thiodiazole-copper.

Meanwhile, the in vitro antifungal activities of the target compounds **4a***–***4r** against *S. sclerotiorum*, *F. oxysporum*, *G. zeae*, *R. solani*, *V. dahlia*, and *B. cinerea* were tested at 50 μg/mL by the mycelial growth rate method [29] and the results were statistically analyzed and listed in Table 3. As shown in Table 3, bioassay results showed that all the target compounds revealed lower antifungal activities against *S. sclerotiorum*, *F. oxysporum*, *G. zeae*, *R. solani*, *V. dahlia*, and *B. cinerea* at 50 μg/mL than those of Carbendazim.

### 2.3. Structure-Activity Relationship (SAR) Analysis

As an extension of this approach, the SAR was deduced on the basis of the inhibitory activity values of the antibacterial and antifungal activities shown in Table 1, Table 2 and Table 3. First, compared to the same substituent at R_2_ and R_3_, the presence of the –Cl atom at R_1_ showed better antibacterial and antifungal activities in the order of **4g** > **4a** and **4k** > **4b**. Second, compared to the same substituent at R_1_ and R_2_, the electron drawing group (–F) at R_3_ could cause an increase in the antibacterial and antifungal activities following the order **4c** > **4a** > **4b**, **4i** > **4g** > **4h**, and **4o** > **4m** > **4n**. Third, compared to the same substituent at R_1_ and R_3_, the smaller substituent groups, such as –CH_3_, at the R^2^ could cause an increase in the antibacterial and antifungal activities. The bioactivities of the target compounds followed the order **4a** > **4d**, **4b** > **4e**, and **4c** > **4f**. 

## 3. Materials and Methods

### 3.1. General Information

The melting points were determined by an uncorrected WRX-4 binocular microscope (Shanghai Yice Tech. Instrument Co., Shanghai, China). ^1^H-NMR and ^13^C-NMR spectral analyses were performed on a Bruker DRX-400 NMR spectrometer (Bruker, Rheinstetten, Germany). HRMS data were measured on an Agilent Technologies 6210 LC/MS TOF mass spectrometer (Agilent, Palo Alto, CA, USA). All reagent products from the Chinese Chemical Reagent Company were analytical or chemically pure.

### 3.2. Chemical Synthesis

#### 3.2.1. Preparation Procedure of Intermediates **2** and **3**

A mixture of substituted thiophenols (70 mmol) and a slight excess of maleic anhydride (84 mmol, 1.2 equivalents) in methylbenzene was added to a 250 mL round bottom flask equipped with a magnetic stirrer and reacted at 50 °C for 0.5 h, and then triethylamine (2 drops) was slowly added and stirred at 70 °C for 4 h. The reaction was quenched to room temperature and then the solvent was removed under reduced pressure. After that, the residues were redissolved with dichloromethane (100 mL) with bath ice and then a significant excess of AlCl_3_ (210 mmol, 3 equivalents) was added. The mixture reaction was stirred in an ice bath for 3–4 h. After the reaction was completed, as determined by TLC, the reaction mixture was diluted with dichloromethane (100 mL) and treated with precooled 5% dilute hydrochloric acid (50 mL). The residues were filtrated, dried under vacuum, and recrystallized from ethanol to give intermediate **2**.

A mixture of intermediate **2** (50 mmol), methanol or ethanol (100 mL), and H_2_SO_4_ (4 mmol) were added in a 250 mL round bottom flask and reacted under reflux conditions for 6–8 h. Upon completion of reaction (determined by TLC), the mixture was quenched to room temperature and the precipitated residues were filtrated, dried under vacuum, and recrystallized from methanol to give pure intermediate **3**.

#### 3.2.2. Preparation Procedure of the Target Compound **4a**–**4r**

A mixture of intermediate **3** (10 mmol), substituted benzenesulfonyl hydrazide (12 mmol, 1.2 equivalents), acetic acid (10 mL), and ethanol (10 mL) was added to a 50 mL round bottom flask equipped with a magnetic stirrer, and reacted under reflux conditions for 2–4 h. Upon completion of the reaction (determined by TLC), the mixture was cooled to room temperature and the precipitated residues was dried under a vacuum and recrystallized from ethanol to give the pure target compounds **4a**–**4r**.

### 3.3. Bioactivity Evaluation

#### 3.3.1. Bacterial and Fungal Strains

All bacterial and fungal strains used in this study were provided by Guizhou University, China.

#### 3.3.2. In Vitro Antibacterial Activity Test

All the taget compounds (7.5 mg) were dissolved in 150 μL DMSO and 80 and 40 μL of the mixture solution was poured into two 15 mL centrifuge tubes with 0.1% Tween aqueous solution (4 mL), respectively. Next, 1 mL Tween aqueous solution with the testing compounds was added into the test tubes containing 4 mL nutrient broth (NB) mediums (Solarbio, Beijing, China) to prepare 5 mL test solutions with concentrations of 200 and 100 μg/mL, respectively. Finally, 40 μL of precultured NB mediums containing *Xoo*, *Xoc*, and *Xac*, respectively, were added to the test tubes and incubated at 30 °C and 180 rpm for 24–48 h until the bacteria were incubated on reaching the logarithmic growth phase. DMSO served as the negative control, whereas Thiodiazole copper and Bismerthiazol served as the positive controls. The OD_595_ values of the cultures were monitored on a Multiskan Sky 1530 spectrophotometer (Thermo Scientific, Poland). Three replicates were conducted for each treatment. Inhibition rate *I* (%) is calculated by the following formula (1), where C is the corrected turbidity value of the untreated NB medium and T is the corrected turbidity value of the treated NB medium.
Inhibition rate *I* (%) = (C–T)/C × 100(1)

Based on the preliminary bioassays results, five corresponding concentration gradients were prepared, and the antibacterial activities (expressed by EC_50_) of some of the target compounds against *Xoo*, *Xoc*, and *Xac* were also evaluated and calculated using SPSS 17.0 software (SPSS, Chicago, IL, USA). The experiments were repeated three times for each compound.

#### 3.3.3. In Vitro Antifungal Activity Test

All the taget compounds (5 mg) were dissolved in 1 mL DMSO and then mixed with 90 mL potato dextrose agar (PDA) medium (Solarbio, Beijing, China). After that, the mixed PDA medium were poured into 6 or 9 dishes and then cooled to room temperature to prepare PDA plates. Mycelia dishes of approximately 0.4 cm diameter were cut from culture medium and then picked up with a germfree inoculation needle to the middle of PDA plate aseptically. The inoculated PDA plates were fostered in an incubator at 28 ± 1 °C for 3–4 days. DMSO served as a negative control, whereas Carbendazim acted as a positive control. Three replicates were conducted for each treatment. The inhibition rate *I* (%) were calculated by the formula (2), where C (cm) represents the diameter of fungi growth on an untreated PDA plate, and T (cm) represents the diameter of fungi on the treated PDA plate.
Inhibition rate *I* (%) = [(C−T)/(C−0.4)] × 100(2)

### 3.4. Statistical Analysis

Statistical analysis was conducted by analysis of variance (ANOVA) in SPSS 17.0 software with equal variances assumed (*p* > 0.05) and equal variances not assumed (*p* < 0.05). Asterisk (*) shown in Table 1, Table 2 and Table 3 indicated the inhibition rates of the target compounds with significant difference at *p* < 0.05 compared to those of positive controls (Thiodiazole copper, Bismerthiazol, and Carbendazim). 

## 4. Conclusions

In this study, a total of 16 novel thiochromanone derivatives containing a sulfonyl hydrazone moiety were designed and synthesized. Bioassay results showed that most of the target compounds revealed moderate to good in vitro antibacterial activities against *Xoo*, *Xoc*, and *Xac,* as well as lower in vitro antifungal activities against *S. sclerotiorum*, *F. oxysporum*, *G. zeae*, *R. solani*, *V. dahlia*, and *B. cinerea.* In particular, compound **4i** exhibited the best inhibitory activities against *Xoo*, *Xoc*, and *Xac*, with the EC_50_ values of 8.67, 12.65, and 10.62 μg/mL, which were superior to those of Bismerthiazol and Thiodiazole-copper. The SAR analysis showed that the size and electron-withdrawing property of substituent groups in R_1_, R_2_, and R_3_ is one of the crucial factors to affect the antibacterial and antifungal activities against the tested bacterial and fungal strains used in this study. The SAR study provided a practical tool for guiding the design and synthesis of novel and more promising active small molecules of thiochromanone derivatives containing a sulfonyl hydrazone moiety for controlling plant bacterial and fungal diseases.

## Data Availability

Data present in this study are available on request from the corresponding author.

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
