# Peer review of "Novel Thiochromanone Derivatives Containing a Sulfonyl Hydrazone Moiety: Design, Synthesis, and Bioactivity Evaluation"

_molecules, 2021, doi:10.3390/molecules26102925_

Round 1
Reviewer 1 Report
Molecules (ISSN 1420-3049)
In the paper entitled ‘Novel thiochromanone derivatives containing a sulfonyl hydra- 2 zone moiety: Design, synthesis, and bioactivity evaluation” the authors designed and synthesized a series of novel thiochromanone derivatives containing a sulfonyl hydra- 11 zone moiety. This work is important, as these compounds have atifungal, antibacterial, anticancer, and insecticidal activity. I therefore think that the manuscript is acceptable for publication.
Specific suggestions are:
- Sclerotinia sclerotiorum ( sclerotiorum), Fusarium oxysporum (F. oxysporum), Gibberella zeae (Gibberella zeae), Rhizoctonia solani (R. solani), Verticillium dahlia (V. dahlia), and Botrytis cinerea (B. cinerea) (Lines 17-20; 62-64) should read: Sclerotinia sclerotiorum, Fusarium oxysporum, Gibberella zeae, Rhizoctonia solani, Verticillium dahlia, and Botrytis cinerea.
- The authors should check for grammar errors. For example, “Recently years,” (Line 28).
- The authors need to make sure that all of the necessary data are provided in the Methods and Materials section. For example, they should provide the supplier for the (Line 325) potato dextrose agar (PDA) medium.
- Maybe the authors can explain briefly in the Conclusions how the 16 novel thiochromanone derivatives differ from each other in terms of in vitro antibacterial activities (Lines 330-337).
- How does this study provide a practical tool for guiding the design and synthesis of novel and more promising active small molecules of thiochro- manone derivatives (Lines 335-336).
Author Response
Reviewer 1
- Sclerotinia sclerotiorum ( sclerotiorum), Fusarium oxysporum (F. oxysporum), Gibberella zeae (Gibberella zeae), Rhizoctonia solani (R. solani), Verticillium dahlia (V. dahlia), and Botrytis cinerea (B. cinerea) (Lines 17-20; 62-64) should read: Sclerotinia sclerotiorum, Fusarium oxysporum, Gibberella zeae, Rhizoctonia solani, Verticillium dahlia, and Botrytis cinerea.
Answer: Thank you very much for your comment. In the revised MS, we have changed “Sclerotinia sclerotiorum (S. sclerotiorum), Fusarium oxysporum (F. oxysporum), Gibberella zeae (Gibberella zeae), Rhizoctonia solani (R. solani), Verticillium dahlia (V. dahlia), and Botrytis cinerea (B. cinerea)” to “Sclerotinia sclerotiorum, Fusarium oxysporum, Gibberella zeae, Rhizoctonia solani, Verticillium dahlia, and Botrytis cinerea”.
- The authors should check for grammar errors. For example, “Recently years,” (Line 28).
Answer: Thank you very much for your comment. In the revised MS, we have checked and corrected the grammar errors throughout the MS.
- The authors need to make sure that all of the necessary data are provided in the Methods and Materials section. For example, they should provide the supplier for the (Line 325) potato dextrose agar (PDA) medium.
Answer: Thank you very much for your comment. In the revised MS, we have added the supplier for the PDA and NB mediums.
- Maybe the authors can explain briefly in the Conclusions how the 16 novel thiochromanone derivatives differ from each other in terms of in vitro antibacterial activities (Lines 330-337).
Answer: Thank you very much for your comment. In the revised MS, we have added the explanation in the Conclusions how the 16 novel thiochromanone derivatives differ from each other in terms of in vitro antibacterial activities.
- How does this study provide a practical tool for guiding the design and synthesis of novel and more promising active small molecules of thiochromanone derivatives (Lines 335-336).
Answer: Thank you very much for your comment. The structure-activity relationship (SAR) analysis could provide a practical tool for guiding the design and synthesis of novel and more promising active small molecules of thiochromanone derivatives.
Reviewer 2 Report
The manuscript "Novel thiochromanone derivatives containing a sulfonyl hydrazone moiety: Design, synthesis, and bioactivity evaluation" has interesting subject matter, but contains little research, requires supplementing the description and discussing the results. The manuscript needs strong corrections to be published.
Detailed comments:
line 19-20 - there is no need to provide abbreviated names of microorganisms in the abstract
I propose to transfer the information from line 74-221 to the Supplementary Materials, and provide the most important information from this section in the manuscript.
Table 1, 2 and 3 - no statistical analysis
Table 1 and 2 - provide full names of microorganisms in the header
The results are very inaccurately described
There is no discussion of the results
There is no description of the method for determining the EC50
When conducting such studies, it would be good to determine the MIC, MBC, time-kill curves of each compound for the tested microorganisms
The methodology lacks information on the origin of the tested strains
Conslusion requires editing and reporting the results obtained.
Author Response
Reviewer 2
- line 19-20 - there is no need to provide abbreviated names of microorganisms in the abstract
Answer: Thank you very much for your comment. In the revised MS, we have deleted the abbreviated names of microorganisms in the abstract.
- I propose to transfer the information from line 74-221 to the Supplementary Materials, and provide the most important information from this section in the manuscript.
Answer: Thank you very much for your comment. In the revised MS, the physical characteristics, 1H NMR, 13C NMR, and HRMS data for all the synthesized compounds had been transferred to Supplementary Materials.
- Table 1, 2 and 3 - no statistical analysis.
Answer: Thank you very much for your comment. In the revised MS, we have added the statistical analysis in Tables 1-3.
- Table 1 and 2 - provide full names of microorganisms in the header
Answer: Thank you very much for your comment. In the revised MS, we have provided the full names of microorganisms in the header.
- The results are very inaccurately described.
Answer: Thank you very much for your comment. In the revised MS, we have corrected the results of our paper.
- There is no discussion of the results
Answer: Thank you very much for your comment. Most of the similar synthesis-related papers published in Molecules have no discussion section in the Results and discussion.
- There is no description of the method for determining the EC50
Answer: Thank you very much for your comment. In the revised MS, we have added the description of the method for determining the EC50 values.
- When conducting such studies, it would be good to determine the MIC, MBC, time-kill curves of each compound for the tested microorganisms.
Answer: Thank you very much for your comment. In our next work, we will determine the MIC, MBC, time-kill curves of each compound for the tested microorganisms following your suggestion.
- The methodology lacks information on the origin of the tested strains.
Answer: Thank you very much for your comment. In the revised MS, we have added the information on the origin of the tested strains.
- Conclusion requires editing and reporting the results obtained.
Answer: Thank you very much for your comment. In the revised MS, we have added the main results obtained in the conclusion.
Reviewer 3 Report
The manuscript “Novel thiochromanone derivatives containing a sulfonyl hydrazone moiety: Design, synthesis, and bioactivity evaluation” describes the syntheses of thiochromanone derivatives containing a sulfonyl hydrazone moiety. The antibacterial activities of these substances were evaluated against Xanthomonas oryzae pv. oryzae (Xoo), Xanthomonas oryzae pv. oryzicolaby (Xoc), and Xanthomonas axonopodis pv. citri (Xac). The antifungal activities were evaluated also for these compounds. The manuscript deserves publication after revision. Listed below are some suggestions for the authors to consider. Abstract, Line 11: In this study, a series of novel thiochromanone … Suggestion: A series of novel thiochromanone … (“In this study” is redundant) Abstract, Line 13-16: Bioassay results showed that the target compounds revealed moderate to good antibacterial activities against Xanthomonas oryzae pv. oryzae (Xoo), Xanthomonas oryzae pv. oryzicolaby (Xoc), and Xanthomonas axonopodis pv. citri (Xac), especially, compound 4i had the best inhibitory activity against Xoo, Xoc, and Xac, with the EC50 values of 8.67, 12.65, and 10.62 μg/mL, which were superior to those of Bismerthiazol and Thiodiazole-copper. Suggestion: Bioassay results showed that the target compounds revealed moderate to good antibacterial activities against Xanthomonas oryzae pv. oryzae (Xoo), Xanthomonas oryzae pv. oryzicolaby (Xoc), and Xanthomonas axonopodis pv. citri (Xac). Compound 4i had the best inhibitory activity against Xoo, Xoc, and Xac, with the EC50 values of 8.67, 12.65, and 10.62 μg/mL, which were superior to those of Bismerthiazol and Thiodiazole-copper. Abstract, Line 21-22: To the best of our knowledge, this is the first report on the antibacterial and antifungal activities of this series of novel thiochromanone derivatives containing a sulfonyl hydrazone moiety. Suggestion: For a scientific research paper it is expected to be “novel”. This sentence could be somewhere else in the manuscript, but the abstract should be restricted to describe in general words the main findings. Page 1, Line 28: Recently years … Suggestion: In recently years … Page 1, Line 43: Thiochromanone, a kind of chromophore compounds, … Suggestion: Thiochromanone, a kind of chromophore compound, … Page 2, Lines 47-48: Therefore, due to its excellent features of low toxicity, easy to be synthetic and derived … Suggestion: Therefore, due to its excellent features of low toxicity, easy to be synthetized and derived … Page 2, Lines 51-52: Some compounds with sulfonyl hydrazone structural units have potential various biological activities … Suggestion: Compounds with sulfonyl hydrazone structural units present various biological activities … Page 2, Figure 71: Meanwhile, a singlet at δ 170.06–170.80 ppm indicated the presence of C=O group. Suggestion: The signal around δ= 170.06-170.80 ppm indicated the presence of C=O group. Page 2, Line 72: The 1H NMR, 13C NMR, and HRMS spectrogram … Suggestion: The 1H NMR, 13C NMR, and HRMS spectra … Page 2, 3, etc., Lines 72, 82, 89, etc: Date … Suggestion: Data … Page 3, etc., Lines 79, etc.: δ: 170.66, 161.84, 159.43, etc. Suggestion: δ: 170.6, 161.8, 159.4 (one decimal place for carbon 13 chemical shifts). Page 7, Lines 260-262: First, compared with the same substituent at R2 and R3 substituent groups, the presence of the –Cl group at R1 substituent group showed better in vitro antibacterial and antifungal activities in the order of 4g > 4a and 4k > 4b. Suggestion: First, comparing with the same substituent at R2 and R3, the presence of the –Cl atom at R1 showed better antibacterial and antifungal activities in the order of 4g > 4a and 4k > 4b. Page 7, Lines 263-266: Second, compared with the same substituent at R1 and R2 substituent groups, the electron drawing group (-F) at R3 substituent group could cause an increase in the antibacterial and antifungal activities followed the order 4c > 4a > 4b, 4i > 4g > 4h, and 4o > 4m > 4n. Suggestion: Second, comparing with the same substituent at R1 and R2, the electron drawing group (-F) at R3 could cause an increase in the antibacterial and antifungal activities following the order 4c > 4a > 4b, 4i > 4g > 4h, and 4o > 4m > 4n. Comment: Removing duplicated words helps reading This could be done all over the manuscript. Page 8, Lines 286: The mixture reaction was stirred with bath ice for 3−4 h. Suggestion: The reaction mixture was stirred in an ice bath for 3−4 h. Page 8, Lines 287: … the mixture reaction … Suggestion: … the reaction mixture…Author Response
Reviewer 3
- Abstract, Line 11: In this study, a series of novel thiochromanone … Suggestion: A series of novel thiochromanone … (“In this study” is redundant) Abstract, Line 13-16: Bioassay results showed that the target compounds revealed moderate to good antibacterial activities against Xanthomonas oryzae pv. oryzae (Xoo), Xanthomonas oryzae pv. oryzicolaby (Xoc), and Xanthomonas axonopodis pv. citri (Xac), especially, compound 4i had the best inhibitory activity against Xoo, Xoc, and Xac, with the EC50 values of 8.67, 12.65, and 10.62 μg/mL, which were superior to those of Bismerthiazol and Thiodiazole-copper. Suggestion: Bioassay results showed that the target compounds revealed moderate to good antibacterial activities against Xanthomonas oryzae pv. oryzae (Xoo), Xanthomonas oryzae pv. oryzicolaby (Xoc), and Xanthomonas axonopodis pv. citri (Xac). Compound 4i had the best inhibitory activity against Xoo, Xoc, and Xac, with the EC50 values of 8.67, 12.65, and 10.62 μg/mL, which were superior to those of Bismerthiazol and Thiodiazole-copper. Abstract, Line 21-22: To the best of our knowledge, this is the first report on the antibacterial and antifungal activities of this series of novel thiochromanone derivatives containing a sulfonyl hydrazone moiety. Suggestion: For a scientific research paper it is expected to be “novel”. This sentence could be somewhere else in the manuscript, but the abstract should be restricted to describe in general words the main findings. Page 1, Line 28: Recently years … Suggestion: In recently years … Page 1, Line 43: Thiochromanone, a kind of chromophore compounds, … Suggestion: Thiochromanone, a kind of chromophore compound, … Page 2, Lines 47-48: Therefore, due to its excellent features of low toxicity, easy to be synthetic and derived … Suggestion: Therefore, due to its excellent features of low toxicity, easy to be synthetized and derived … Page 2, Lines 51-52: Some compounds with sulfonyl hydrazone structural units have potential various biological activities … Suggestion: Compounds with sulfonyl hydrazone structural units present various biological activities … Page 2, Figure 71: Meanwhile, a singlet at δ 170.06–170.80 ppm indicated the presence of C=O group. Suggestion: The signal around δ=06-170.80 ppm indicated the presence of C=O group. Page 2, Line 72: The 1H NMR, 13C NMR, and HRMS spectrogram … Suggestion: The 1H NMR, 13C NMR, and HRMS spectra … Page 2, 3, etc., Lines 72, 82, 89, etc: Date … Suggestion: Data … Page 3, etc., Lines 79, etc.: δ: 170.66, 161.84, 159.43, etc. Suggestion: δ: 170.6, 161.8, 159.4 (one decimal place for carbon 13 chemical shifts). Page 7, Lines 260-262: First, compared with the same substituent at R2 and R3 substituent groups, the presence of the –Cl group at R1 substituent group showed better in vitro antibacterial and antifungal activities in the order of 4g > 4a and 4k > 4b. Suggestion: First, comparing with the same substituent at R2 and R3, the presence of the –Cl atom at R1 showed better antibacterial and antifungal activities in the order of 4g > 4a and 4k > 4b. Page 7, Lines 263-266: Second, compared with the same substituent at R1 and R2 substituent groups, the electron drawing group (-F) at R3 substituent group could cause an increase in the antibacterial and antifungal activities followed the order 4c > 4a > 4b, 4i > 4g > 4h, and 4o > 4m > 4n. Suggestion: Second, comparing with the same substituent at R1 and R2, the electron drawing group (-F) at R3 could cause an increase in the antibacterial and antifungal activities following the order 4c > 4a > 4b, 4i > 4g > 4h, and 4o > 4m > 4n. Comment: Removing duplicated words helps reading This could be done all over the manuscript. Page 8, Lines 286: The mixture reaction was stirred with bath ice for 3−4 h. Suggestion: The reaction mixture was stirred in an ice bath for 3−4 h. Page 8, Lines 287: … the mixture reaction … Suggestion: … the reaction mixture…
Answer: Thank you very much for your comments. In the revised MS, we have corrected the MS following your comments.
Reviewer 4 Report
After reading this submission just to revise minor types of English grammar
Author Response
- After reading this submission just to revise minor types of English grammar.
Answer: Thank you very much for your comment. In the revised MS, we have corrected the MS following your comment.
Round 2
Reviewer 2 Report
There is still no statistical analysis (statistical tests must be applied) in the tables and the results should be described after the statistical analysis has been performed.
Author Response
There is still no statistical analysis (statistical tests must be applied) in the tables and the results should be described after the statistical analysis has been performed.
Answer: Thank you for your comment. In the revised MS, we have added the statistical analysis in Tables 1-3.
Reviewer 3 Report
The manuscript “Novel thiochromanone derivatives containing a sulfonyl hydrazone moiety: Design, synthesis, and bioactivity evaluation” describes the syntheses of thiochromanone derivatives containing a sulfonyl hydrazone moiety. The antibacterial activities of these substances were evaluated against Xanthomonas oryzae pv. oryzae (Xoo), Xanthomonas oryzae pv. oryzicolaby (Xoc), and Xanthomonas axonopodis pv. citri (Xac). The antifungal activities were evaluated also for these compounds. The authors improved the overall quality of the manuscript. The manuscript deserves publication after revision. Listed below are some suggestions for the authors to consider.
Page 1, Line 25: Recently years …
Suggestion: In recently years …
Page 2, Line 67: The singlet around δ = 170.06–170.80 ppm indicated the presence of C=O group.
Suggestion: The signal in δ = 170.0–170.8 ppm indicated the presence of C=O group.
Scheme 1. Synthetic route of the target compounds 4a−4r.
Suggestion: Scheme 1. Synthesis of compounds 4a−4r.
Page 2, Line 74: Table 1 showed that most of the target compounds …
Suggestion: Most of the target compounds …
Author Response
- Page 1, Line 25: Recently years …
Suggestion: In recently years …
Answer: Thank you for your comment. In the revised MS, we have changed “Recently years …” to “In recently years …”
- Page 2, Line 67: The singlet around δ = 170.06–170.80 ppm indicated the presence of C=O group.
Suggestion: The signal in δ = 170.0–170.8 ppm indicated the presence of C=O group.
Answer: Thank you for your comment. In the revised MS, we have changed “The singlet around δ = 170.06–170.80 ppm indicated the presence of C=O group.” to “The signal in δ = 170.0–170.8 ppm indicated the presence of C=O group.”
- Scheme 1. Synthetic route of the target compounds 4a−4r.
Suggestion: Scheme 1. Synthesis of compounds 4a−4r.
Answer: Thank you for your comment. In the revised MS, we have changed “Scheme 1. Synthetic route of the target compounds 4a−4r.” to “Scheme 1. Synthesis of compounds 4a−4r.”
- Page 2, Line 74: Table 1 showed that most of the target compounds …
Suggestion: Most of the target compounds …
Answer: Thank you for your comment. In the revised MS, we have changed “Table 1 showed that most of the target compounds …” to “Most of the target compounds …”